# On the Benefits of Learning to Route in Mixture-of-Experts Models

**Nishanth Dikkala**
Google Research

**Nikhil Ghosh**
UC Berkeley *

**Raghu Meka**
UCLA *

**Rina Panigrahy**
Google Research

**Nikhil Vyas**
Harvard University *

**Xin Wang**
Google Research

## Abstract

Mixture-of-Expert (MoE) models, such as the Switch Transformer, allow us to scale model sizes while keeping the amount of compute time fixed. Prior work has established the computational benefits of MoE models. We investigate whether they offer benefits other than scaling up. A core component of these models is a router that routes input tokens to different experts in a layer. We show theoretical and empirical evidence that the router's ability to route intelligently confers a significant advantage to MoE models. We study synthetic settings where the input data is distributed in clusters and show theoretically and empirically that the router learns the cluster structure. Then we perform experiments on real data using the T5X library, where we observe that a trainable router confers a non-trivial benefit instead of a non-trainable router.

## 1 Introduction

With the advent of large Transformer based models (Vaswani et al., 2017; Kaplan et al., 2020; Brown et al., 2020), model size has increasingly become an issue with neural networks. Larger models continue to improve quality; however, this has started to become very expensive. A promising idea to scale up model size without increasing compute time is that of *conditional compute architectures* which activate only a part of the network on any example. Conditional compute architectures hold significant potential for saving compute as not all model parameters are used in every example. One of the most successful realizations of this idea is the Switch Transformer (Fedus et al., 2022b) which uses a *Mixture-of-Experts* (MoE) architecture. Mixture-of-Expert models (Figure 1) in the simplest and most-often used setting are described by a set of *experts* $E_1, \dots, E_N$ and a routing function (aka *gating mechanism*). Each expert is itself

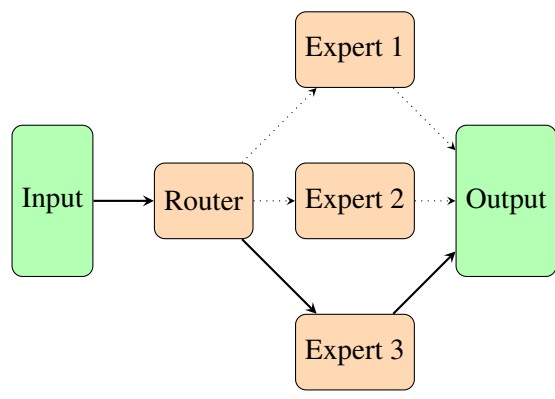

Figure 1: A schematic diagram showing the abstract Mixture-of-Experts architecture for Neural Networks.

a small neural network (a popular choice for the expert architecture is 2 fully connected layers with a non-linearity in between). The routing mechanism is described by a trainable weight matrix $W$ and routes each input $x$ to an expert (or a few experts) based on the soft-max function applied to $W \cdot x$. The core principle behind using MoE models is that only one (or a few) of the $E_i$'s is invoked per example. Thus, when normalizing for compute time, the learned model can draw from the benefits of having many more parameters than a *dense model* (which can be viewed as having a single expert).

In this work, we look closely at a core component of MoE models: the router, which routes different inputs to different experts. By design, MoE models confer computational advantages, and several works (Fedus et al., 2022b; Riquelme et al., 2021; Fedus et al., 2022a) have demonstrated that this speed-up is achieved without any loss in accuracy. We outline two major reasons for this advantage:

- First, an MoE model using the same compute as a dense model has a larger number of effective parameters, and hence can use different sets of parameters for different *regions* in the

---

Work done while at Google Research

input space. Such an advantage would also hold, for example, in a neural network that uses a fixed hash function to assign the input to an expert. Note here that the assignment of inputs to experts need not be learned.

- The second and more interesting reason is that the router is learning an intelligent routing function that cleverly partitions the input space amongst the experts. An example would be a router that routes to experts based on the clusters in the input i.e., all examples in a cluster are routed to a single expert.

We seek to theoretically and empirically understand what role the router plays in delivering the advantages that MoE models enjoy over dense models. Specifically, we study the following questions about the routing mechanisms in MoE models:

- Can the router learn intelligent and interpretable routing functions? For instance, can the router identify any latent clusters in the data?

- How important is the learnability of the routing mechanism as opposed to routing based on a fixed data-independent router (for instance, using a random locally-sensitive hash function)?

- Can we prove that MoE models achieve good accuracy in well-studied clusterable settings such as mixtures of Gaussians?

Zoph et al. (2022a) and others have tried to identify patterns in what the router is learning and have uncovered some insights. However, they are hard to interpret due to the complexity of real data. Here, we study MoE models on various synthetic tasks, which sheds clearer light on the role of the router. To begin with, we look at tasks where the input has a clusterable structure, such as a mixture distribution over separated Gaussians, and study whether the router can learn to route inputs according to the cluster identity. We show theoretical and empirical results in the affirmative for the router learning the underlying cluster structure of the data. We also perform experiments comparing MoE models with a trainable vs. a non-trainable router on the Common Crawl (mC4) dataset and observe that a learned routing function performs significantly better than a routing function decided in a data-independent manner.

## 1.1 Our Empirical Results

Several works (see Section 1.3 and references therein) have empirically demonstrated that MoE models exhibit significant advantages over dense models for very large models. We aim to understand this phenomenon further, identify the benefits (if any) of learned routing vs. fixed routing, and determine whether routers trained with gradient descent identify latent structures in data. An important design choice for MoE models is the router architecture. We use a Softmax-layer router which is the most popular choice for MoE models (Fedus et al., 2021). The routing mechanism is described by a trainable weight matrix $W$ and routes each input $x$ to an expert (or a few experts) based on the soft-max function applied to $W \cdot x$. There are more recent variants such as Zoph et al. (2022b); Zhou et al. (2022); Puigcerver et al. (2023) which perform better than Switch Transformer but all of these works use the same Softmax-layer router as a backbone. We do note that other routing mechanisms which do not use a Softmax-layer have been studied (See Section 1.3) but we do not study those in our work. For interpretability and to identify the more fine points we seek, we introduce the study of the performance of MoE models under various synthetic problems. Most of these problems are quite well-studied in clustering and unsupervised learning and are independently interesting. We describe these settings first:

1. **Mixtures of Gaussians**: One of the most studied models for clusterable data is mixtures of Gaussians. We study the performance of MoE models when the input examples come from a mixture of Gaussians and the labels are specific to each cluster (either constant or linear functions).

2. **Mixtures of Subspaces**: We move from Mixtures of Gaussians to Mixtures of subspaces, this is a setup in which the input clusters are not necessarily separable in Euclidean distance.

3. **Tokenized mixtures of Gaussians**: The Transformer architecture which MoE models try to improve on is a sequence prediction model. Consequently, an important design feature of many MoE architectures is that one routes each token in the sequence independently of the other tokens.

   To capture this sequence setting, we consider a setup where we have a sequence of tokens

with each token being sampled from a mixture of Gaussians.

4. **Dictionary Learning**: This is a more complicated set up where each input is created using combined information from 2 or more clusters. This helps us model practical data more accurately as there is some amount of structured overlap between concepts here. Here, although we perform better than dense models in certain settings, we do not observe intelligent routing as a frozen router performs just as well as learnable router.

5. **CIFAR-100 class clusters**: In this setting we consider clusters defined by CIFAR-100 classes. The output is a linear function per cluster.

Finally, we perform some experiments on real data. We pre-train T5 transformer architectures (Raffel et al., 2020) and measure the pre-training accuracy. We compare three different models: (i) a dense 12-layer T5 model, (ii) a 12-layer MoE model with three sparse layers and top-2 routing and a trainable router, (iii) a 12-layer MoE model with three sparse layers and top-2 routing and a fixed non-trainable router.

In the above settings, we have two main results corresponding to our questions on understanding the routing behavior. For details on the results and the experimental methodology in Section 3.

**Routers learn to route intelligently based on input structure:** In our empirical experiments, we find that routers of MoE models can adapt to the input structure. Specifically,

- In Mixtures of Gaussians experiments, with constant or linear outputs per Gaussian cluster, we find that routers learn to map clusters to experts. That is, nearly all examples from a cluster are routed to the same expert (Figures 2(c), 3(b)).

- When trained on Mixtures of Gaussians with spurious dimensions, i.e., dimensions along which the marginal of all Gaussian clusters is the same, we find that the router learns to ignore these directions (See the last paragraph of Section 3.1.

**Learning to route offers an advantage to MoE models:** In all of our settings described above, we universally find that learned routers significantly outperform MoE models with random routers. We study these effects by plotting the data scaling

curves (Figures 2(a), 3(a) 4 for Mixtures of Gaussians and Figure 6 for Mixtures of Subspaces) as well as pointwise comparisons for the CIFAR-100 based task and the T5X task.

## 1.2 Our Theoretical Results

We seek to show that MoE models provably achieve good performance (and intelligent routing) in classically well-studied instances of clusterable distributions. To our knowledge, such results are not known[1]. As a first step, in this direction, we study simple models with a single router that routes inputs to experts, and each expert either applies a linear function on the input routed to it or outputs a fixed vector. We also analyze convergence under gradient descent with population gradients to minimize technical machinery and highlight the important points. The arguments would extend naturally to sample gradients as long as we have enough samples by standard concentration inequalities.

In our first theoretical result, Theorem 1, we show that the router can identify relevant directions and remove spurious directions. Concretely, we show that if we have $d$-dimensional inputs that are clusterable in a subspace of dimension $d' \ll d$ (unknown to us) and has independent spherical Gaussian noise along the other $d - d'$ directions, the router aligns itself with the hidden $d'$-dimensional subspace. This tells us that the router can perform intelligent projection operations. We prove this result more generally for any neural network architectures with a regularization term for the bottom-layer weights, which could be of interest independently. The proof exploits rotational symmetry and *self-decomposability* of the Gaussian distribution.

Next, we study the setting of a mixture of $k$ Gaussians. We assume that the $k$ Gaussians are spherical and live in a well-separated high-dimensional space where the notion of well-separability is analogous to the classical notion in Dasgupta and Schulman (2007). The second assumption we make is in the initialization of the weights of the router matrix. Rather than randomly initializing them in a data-independent manner, we set each row of the router as a randomly sampled training example. This is similar in spirit to Dasgupta and Schulman (2007).

Under these assumptions, we show that each

---

[1]As described in Section 1.3, Chen et al. (2022) recently showed an elegant provable guarantees for MoE models. However, their result applies to a data distribution they define with a specific purpose and perhaps does not correspond to an independently interesting clusterable instance

expert specializes to a specific cluster center. By using $\Theta(k \log k)$ experts, we can ensure that every cluster has at least one expert specializing in that cluster. By choosing the router learning rate appropriately, we can also argue that while training the router, weights do not change too much, while the experts specialize in their clusters. Our result is given in Theorem 2.

We also show a result similar to the above for mixture of two Gaussians with no assumption on the initialization in Appendix F.

### 1.3 Prior work

Mixture-of-Expert models (Jacobs et al., 1991; Shazeer et al., 2017; Fedus et al., 2022b; Riquelme et al., 2021) have been used in various forms in machine learning over the last three decades. We refer to the excellent survey of Fedus et al. (2022a) for their history. In our work, we focus specifically on their use in neural networks. We describe the most relevant works to ours below.

Roller et al. (2021) studied routing based on Hashing the input tokens. This is an example of a routing scheme where the assignment of inputs to experts is not learned but is not necessarily the strongest such routing. Clark et al. (2022) showed that such a routing scheme does not scale as well as architectures with trainable routers. We consider a more general setup where the routing is not fixed at initialization, but the ability to learn to route is limited. We show that even in this setting, learning to route is essential.

Recent works of Chen et al. (2022) and Baykal et al. (2022) study the theoretical benefits of MoE models over dense models. Similar to us, Chen et al. (2022) focus on problems where the input structure has clusters. We differ from this work in our focus on more elaborate synthetic and real-world examples. We give a more in depth comparison of our results with those of Chen et al. (2022) along with a discussion of additional related work in Appendix B.

**Remark 1.** *In trying to compare the performance of MoE models with dense models, it is important to understand the difference between over-parameterized and under-parameterized settings[2]. In under-parameterized settings, irrespective of the router learning intelligent routing, MoE models have a clear advantage over dense models as they simply have more parameters. In an over-*

parameterized model, the benefit of MoE models having more parameters goes away. Still, one could hope that MoE models outperform dense models, particularly when inputs are clustered, as they might have a better implicit bias for such inputs. We don't observe this in our experiments in Figure 2(a) where the largest dense model outperforms the MoE model even though both are over-parameterized.

## 2 Preliminaries

We use $[n]$ to denote the list of integers from $[1, 2, \ldots, n]$. We denote the probability distribution of a $d$-dimensional Gaussian with mean vector $\mu$ and covariance $\Sigma$ by $\mathcal{N}(\mu, \Sigma)$. In our work, we will deal with spherical Gaussians where the covariance matrix $\Sigma = \sigma I_d$. Given two $d$-dimensional spherical Gaussians $\mathcal{N}(\mu_1, \sigma_1 I_d), \mathcal{N}(\mu_2, \sigma_2 I_d)$, we say they are $c$-separated if $\|\mu_1 - \mu_2\|_2 \geq c \max\{\sigma_1, \sigma_2\}\sqrt{d}$ (from Dasgupta and Schulman (2007)).

The neural networks we use in our synthetic experiments have the following form: Every input is first processed by the router and assigned to a single expert, and the output of this specific expert is the neural network's output. We describe MoE models by specifying the router and the experts. For example, a linear router with constant experts refers to a trained model which

1. On input $x$, computes $Wx$ where $W$ denotes the router weights. The input is assigned the expert $j$ where $j = \arg\max_i (Wx)_i$ i.e. $j^{\text{th}}$ entry of $Wx$ is the largest entry.

2. The expert $j$ outputs a learned vector $w_j$ and hence $w_j$ is the output[3] of the neural network.

We note that this description applies to trained models. While training, we do a weighted combination of experts instead of routing to the top expert. All our results are for a trained network for which we do top-1 routing. See Appendix C for more details.

### 2.1 Metrics

We will use various metrics in our synthetic experiments to understand the behaviour of MoE models. Here we list these metrics:

---

[2]We define these settings based on whether train loss can reach near 0 or not respectively

[3]The output of the expert in this case does not depend on the input as the expert is a "constant" expert. By "constant" we mean independent of input, not that it is fixed throughout training, i.e., $w_j$ is still learned. If we had a linear expert with weight $w_j$, the output would be $w_j^T x$.

- **Test Loss:** The loss of the MoE model on the underlying distribution. If test loss tends to 0 we are able to argue that gradient descent can find a network which can represent the true labelling function.

- **Average Sparsity per Cluster:** MoE routers typically output a probability distribution over the experts indicating which experts' to rely on for the output. Our input data will usually be a uniform mixture of $k$ underlying (clustered) distributions $C_1, C_2, \ldots, C_k$. We want to understand how well the router is able to "discover" these clusters. To do this, we define a notion of 'sparsity' which is a continuous quantity which captures how sparse the probability vector output by the router is. We consider the following quantity:

$$\mathbb{E}_{i \in [k]} H(\mathbb{E}_{x \in C_i} R(x))$$

where $H$ is the entropy function and $R(x)$ is the vector of assigned probabilities to each expert by the router $R$ on input $x$. Intuitively, we take the average (over a cluster) of the assignment to experts and calculate the entropy and then we also average over all the clusters. A smaller average sparsity metric across inputs from a cluster indicates that all the inputs from a cluster are being routed to a small set of experts. A similar metric is used in Chen et al. (2022) as well.

- **Average Sparsity per Cluster (Shuffled Router):** Suppose $R$ is a linear router i.e. $R(x) \coloneqq \mathrm{softmax}(Wx)$ where $W \in \mathbb{R}^{\#\text{experts} \times \text{input dimension}}$. Let $W'$ be a random shuffling of the columns of $W$. Let $R'$ denote the router obtained by using $W'$ instead of $W$. This metric is the quantity $\mathbb{E}_{i \in [k]} H(\mathbb{E}_{x \in C_i} R'(x))$. This is the exact same quantity as the previous metric except that we use the shuffled router $R'$ instead of $R$. We use this metric as a baseline for the previous metric. This serves as a good baseline as various statistics such as average $L_p$ norm of entries are shared between $W$ and $W'$.

## 3   Experimental Results

We present our experiments on synthetic (Sections 3.1-3.6) and real datasets (Section 3.7) in this section. Full details of our experimental setup are in Appendix C.

### 3.1   Mixture-of-Gaussians inputs and constant vector output per Gaussian

This is the simplest setting we consider. The input is a mixture of well-separated Gaussians, and the output is a random constant vector per Gaussian. Note that an MoE model can represent this label function with a linear router and "constant" experts, i.e., each expert outputs a fixed albeit learned vector independent of the input.

**Can gradient descent discover underlying clusters?** In our first experiment (Figure 2), we set the dimension of the input to be 24 and take the input distribution to be a uniform mixture of 64 well-separated spherical Gaussians. The output is a randomly sampled 10-dimensional vector per Gaussian cluster. We use an MoE model with a linear router and "constant" experts. See Appendix C for more details. In Figure 2(a), we see that as the number of samples increases, the loss tends to 0. A similar effect is seen in Figure 2(b) where we run the same experiment but vary the number of experts (the number of samples is set to a very large value). When the number of experts is less than the number of clusters (64) we see that the performance cannot achieve 0 test loss. This is expected as the architecture is unable to represent the actual function. We see that when the number of experts equals the number of clusters (64), the loss is near 0. This is surprising in comparison to our theoretical result (Theorem 2), where we need an additional $\log(\#clusters)$ factor to prove convergence to a good solution. A similar logarithmic factor is incurred in the analyses of Dasgupta and Schulman (2007), Chen et al. (2022).

Next, we explore the behavior of the router. In Figure 2(c) we see how sparsity (average sparsity per cluster, see Section 2.1) changes with increase in training data size. We find that sparsity tends to 1 with increasing dataset size. Hence each cluster gets mapped to a single expert. Comparison with the shuffled router, which has much higher sparsity, shows this is not due to spurious factors such as large norm router weights.

**Benefits of learning to route** Our next question is whether a trainable router helps the performance of MoE models. To do this, we can compare the performance of an MoE model with a frozen router. But with a frozen router, an MoE model with a linear router and "constant" experts will not be able to reach a train loss of 0. To rectify this, we move to an MoE model with depth-3 experts and compare

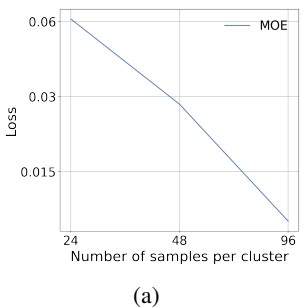 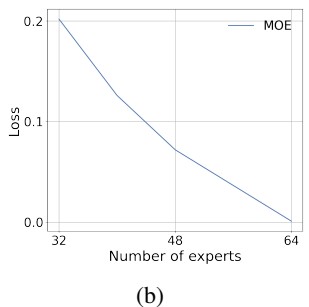 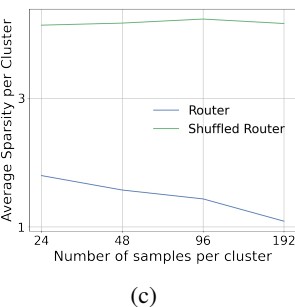

|     |     |     |
| --- | --- | --- |
| (a) | (b) | (c) |

Figure 2: **Mixture-of-Gaussians inputs and constant vector output per Gaussian.** We use a neural network with a linear router, and experts that output a constant (learned) vector. In Figure 2(a), we take the number of experts equal to the number of clusters and see that as the number of samples increases, the loss tends to 0. In Figure 2(b), we set the number of samples to be very large and see that as soon the number of experts matches the number of clusters (64), the loss drops to near 0. Finally, in Figure 2(c), we examine the learned router and find that the router learns to map each cluster to a unique expert while a shuffled router does not.

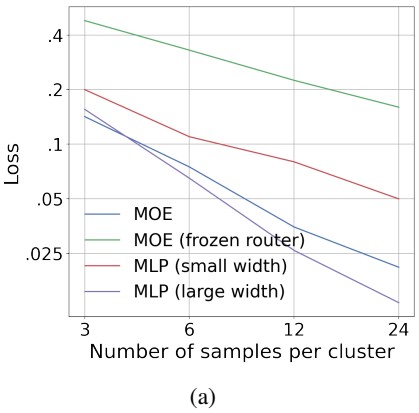 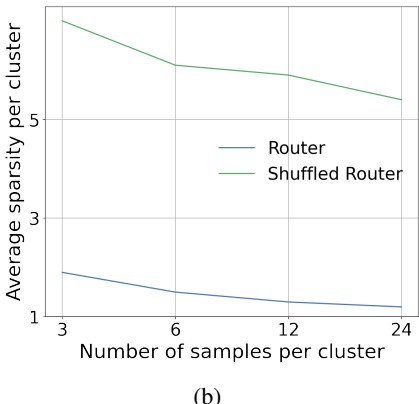

|     |     |
| --- | --- |
| (a) | (b) |

Figure 3: **Mixture-of-Gaussians inputs and constant vector output per Gaussian** We use a neural network with a single layer router i.e. a linear router and depth-3 MLPs as experts. In Figure 3(a) we take number of experts to be equal to the number of Gaussian cluster and we see that the as the number of samples increases the loss tends to 0 while the performance of the model with frozen router is much worse. In Figure 3(b) we examine the learned router and find that the router learns to map Gaussian clusters to unique experts while shuffled router is not able to do so.

with the same MoE model with depth-3 experts but with a frozen router. Figure 3(a) shows that the MoE model with a learned router outperforms the MoE model with a frozen router. Note that the training loss of all models is near zero; thus, this is a gap in generalization than in training error. In Figure 3(b), we again see similar behavior to what we saw in Figure 2(c). See Appendix C.1 for a related experiment.

**Comparison to Dense models:** Figure 3(a) also compares MoE models with two dense MLP networks. One matches the number of parameters of a single expert (so compute time is similar to MoE), and another matches the total number of parameters of the entire MoE model (much more expensive computationally). As MoE models have one routing layer, for a fair comparison, we allow

MLP models to have one more depth than the MoE models. We find that the MoE model outperforms the smaller MLP but slightly underperforms the larger MLP model.

**Identifying non-spurious dimensions:** We repeat the same experiment with $64$ clusters, but this time we embed these in $240$ dimensions, out of which $240 - 24 = 216$ are spurious dimensions, i.e., they are i.i.d. Gaussians for all clusters. We again find that training with this setup leads to a small test loss ($0.01$, $1$ being trivial, and $0$ being perfect). This indicates that gradient descent converges to a near-perfect solution. Let $w_i$ denote the weights of the router corresponding to expert $i$ and $w_i'$ denote the projection of $w_i$ on the $24$ non-spurious dimensions. We find that the quan-

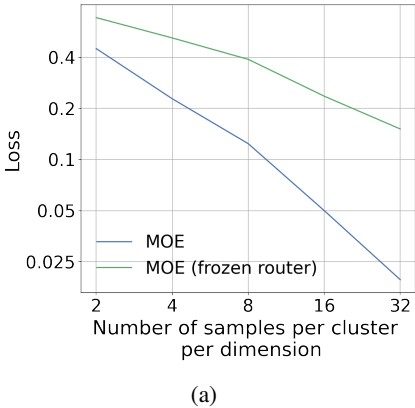

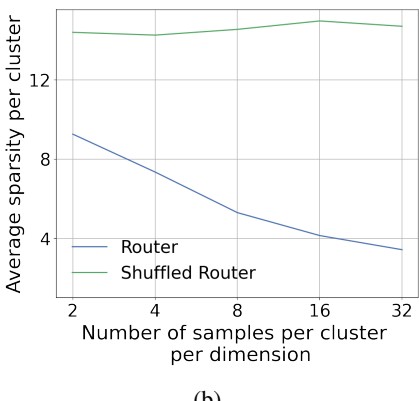

|(a)|(b)|

Figure 4: **Mixture-of-Gaussians inputs and linear function per Gaussian** We use a neural network with a single layer router, i.e., a linear router, and depth-3 MLPs as experts. In Figure 4(a), we see that as the number of samples increases, the loss tends to 0, and the model's performance with a frozen router is much worse. In Figure 4(b), we examine the learned router and find that the router learns to map Gaussian clusters to a small set of experts while the shuffled router does not.

tity $\frac{\sum_i \|w_i'\|^2}{\sum_i \|w_i\|^2}$ which intuitively denotes how much the router focuses on the non-spurious dimensions, goes from $.1 = 24/240$ at initialization to $.84$ after training without weight decay. Adding weight decay increases it to $.9$ without loss in performance. Hence the router learns to focus mostly on the 24 non-spurious dimensions out of the 240 dimensions. This connects with our Theorem 1, which predicts this behavior.

### 3.2 Mixture-of-Gaussians inputs and linear function per cluster

We next generalize the previous setting. The input distribution remains the same but has a separate linear function for each cluster. For each Gaussian $N(\mu, I_d)$, we now sample a random vector $w$ linear function $w$, and on input $x$ from this Gaussian, the output is $w^T(x - \mu)$ [4]. We train an MoE model with a linear router and depth-3 MLP experts on this task. In Figure 4(a), we observe that the loss goes down as we increase the number of samples per cluster so that the network can learn this function. In Figure 4(b), we analyze what the router learns. To do this, we plot the average sparsity per cluster as we increase the number of samples. We see that the sparsity decreases as the number of samples increases. As a baseline, we plot the same quantity after shuffling the router weights and observe no improvement in sparsity in this setting. This demonstrates that the router routes according to the latent clusters much better than the shuffled router.

### 3.3 Non-linear clusters: Mixture-of-subspaces

In this setting, the input is a mixture of $k$ randomly chosen rank-$t$ subspaces with added noise to samples. We defer more details on the setting to Appendix C. In Figure 6, we compare the MoE model with a learned router to one with a frozen router. We find that learning to route is important for the performance for a large range of dataset sizes.

### 3.4 CIFAR-100 class clusters:

We consider the following regression task. We construct a dataset where each input is a pair of the form (CIFAR-100 image, a random vector $v$). The ground truth target output is defined as $w_i^T v$ where $i$ is the class of the image, and $w_i$ is a class-specific random vector. The model we use is of the following form: we have a CNN router and depth-3 MLP as experts. We set the number of experts to 200 (Theorem 2 suggests that for $k$ clusters we need $O(k \log k)$ experts). The CNN is only given the image as input, and experts are given the vector $v$ and the penultimate layer[5] of the CNN. The loss when we allow the CNN router to learn is $.55$, while it is $.80$ when we freeze it.

### 3.5 Sequence Data:

One difference between real transformer models and our experiments above is that we were not processing sequence data where the input is split into tokens. We also perform experiments which

---

[4]We center the linear function as else, the values become close to constant per cluster again.

[5]This *helps* the model in the case we freeze the router by making sure that experts also get information about the image.

demonstrate that our insights can be extended to when the input is given as a sequence of tokens. See Appendix C.2 for more details in this setting.

### 3.6 A Dictionary Learning Inspired Setting

We now explored whether MoE can route in a more complex setting. Taking inspiration from dictionary learning, we create an experiment where there are $k$ atoms $\{\mu_i\}_{i=1}^k \in \mathbb{R}^d$. Each of these atoms corresponds to a cluster. An input $x$ is generated by selecting a few of these atoms (2 or 3 in our experiments), sampling Gaussian random variables centered at each of the selected atoms, and finally adding up the sampled variables. The target output $y$ is a scalar obtained by multiplying $\mu_i$ with an atom-specific vector $w_i$ for the selected atoms and adding up the results. Depending on the problem parameters, this problem can be easy or hard. Although we observed the MoE model achieve a better test loss than dense models (0.19 vs 0.03) in some settings, we did not observe sparsity in routing similar to what we saw in previous sections. Moreover, the test loss of 0.03 was also achieved when the router was frozen at initialization. In addition, the test loss was not very close to 0 indicating that this setting is more challenging for MoE routers to learn the latent structure. See Appendix C.3 for more details.

### 3.7 Language Modeling on mC4

One of our aims is the study the benefit of learning to route in MoE models on real data. We study language modeling objective on the Common Crawl (mC4) dataset with T5 transformers. A first approach one could consider would be to compare neural networks where the router is learned versus networks where the router is frozen. Unfortunately, this cannot be directly achieved in standard Transformer-based MoE models as the router is also used between layers. Hence even if we were to freeze the router itself, the earlier layers could learn to do the routing implicitly.

To get around this, we a) reduce the model dimension to 32 and b) freeze the router. We do (a) to reduce previous layers' capacity to learn to route. Note that there could still be some learning to route happening implicitly in the earlier layers. Hence, the performance of this model will be an upper bound on the performance of a model which does not learn to route. With these changes, we first find that an MoE model with a frozen router significantly outperforms a dense model with similar FLOPs. More relevant to our study, the MoE

model with a frozen router is significantly outperformed by an MoE model with a trainable router. The performance of these three models is compared in Figure 5. We note that our artificial constraint on the model dimension makes it hard to directly draw conclusions for the setting of real-world Transformer models. However, it is otherwise challenging to disentangle implicit router learning by the pre-router layers in a large model from the actual role of allowing a router to train. See Appendix C.4 for experiments at additional scales with T5 models.

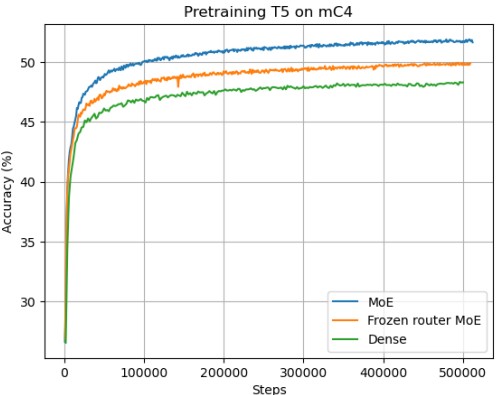

Figure 5: Pretraining Accuracy on mC4. Each run is a 12 layer model with model dimension reduced to 32 (from 768). This has been done to better study the role of learning to route. See Section 3.7.

## 4 Theoretical Results

We state and describe our theoretical results in this section deferring full proofs to the Appendix. Our first result proves that MoE routers can identify relevant directions and remove spurious directions. We prove this result for any MoE architecture which passes the input through a softmax router first with a regularization term for the router weights.

### 4.1 Router Learns to Ignore Spurious Directions

We consider a setting where we are given inputs from a mixture of $k$ spherical Gaussians in $\mathbb{R}^d$. The Gaussians are well separated in a dimension of rank $z$, but there is no clustering when projected to other $d-z$ dimensions. The ground truth is a fixed vector $v_c$ corresponding to the Gaussian $x$ is drawn from.

**Theorem 1.** *(Informal) Given a target function $y(x)$ which depends on only a projection of $x \in \mathbb{R}^d$ to $d' < d$ dimensional subspace $V$ and a prediction function $f$ which is of the form $g_U(Wx + b)$ for some function $g$ (parameterized by $U$), any local*

*minimum of the regularized loss $\ell(y, f)$ must have each row of $W$ lie within $V$.*

This shows us that routers indeed have the capacity to learn to filter out *signal* directions from *noise* directions *even* when the final function being computed is a non-linear function of the input.

Next, we show that routers can learn latent clusters when inputs come from a mixture of Gaussians. We choose mixture of Gaussians as it is known to be a highly powerful modeling framework in practice (Khansari-Zadeh and Billard, 2011; Varadarajan et al., 2013) while also theoretically capable of representing arbitrary distributions using Gaussians as *basis* functions. They are also well-studied and are known to be amenable for theoretical analysis.

### 4.2 Router Learning Latent Clusters for a Mixture of $k$ well-separated Gaussians

We consider a setting with a mixture of $k$ Gaussians. Let $\{g_c = \mathcal{N}(\mu_c, \sigma_c I_d)\}_{c=1}^k$ be the $k$ spherical component Gaussians of a mixture distribution defined by $\sum_{c=1}^k w_c g_c$. We assume that all the mixture weights $w_c$ are uniformly $= 1/k$ for simplicity.

**MoE architecture:** The neural network is of the form: $f(x) = \sum_{i=1}^e p_i u_i$ where $p_i = \text{softmax}(w_i^T x)$ and $u_i, w_i$'s are parameters. The $w_i$'s represent the router and the $u_i$'s represent the experts.

First, we note strong similarities this setting has with the $k$-means problem in Lemma 1. Then we show that if we initialize the router weights in a specific way using the training data, each expert specializes to a specific cluster center. By using $\Theta(k \log k)$ experts, we can ensure that every cluster has at least one expert specializing in that cluster.

**Theorem 2.** *(Informal) Given $n = \Theta(dk \log k)$ samples drawn from a mixture of $k$ spherical Gaussians in $d$-dimensions which are $c$-separated for some constant $c$ as described above, consider an instantiation of the above MoE architecture with $O(k \log k)$ experts. If we initialize the router weights $w_i$ to randomly drawn examples from the train set, the router will learn to route examples according to the cluster they belong to.*

In Theorem 2, we had to initialize our router weights in a smart manner using the inputs to ensure that they identify the latent clusters. Such a utilization of stratified subsampling of a dataset is not uncommon in Machine Learning. It is used in K-means algorithms, for instance. It is also a key component of Curriculum Learning (Bengio et al., 2009). We also have stronger theoretical results

for Mixture of 2 Gaussians which do not need to initialize the router intelligently in Appendix F. In all our analyses we assume a balanced mixture of Gaussians as that simplifies the proof considerably. However, we believe our results of Theorem 2 and 4 can be generalized to hold even when the mixture weights are not uniform.

## 5 Conclusion

We studied the role of learning a router in MoE models and identified a clear advantage in many synthetic and real settings for learning to route. In addition, we also saw that the MoE router is capable of learning latent cluster structure in some scenarios and less capable in more complex scenarios like the dictionary learning setting. We believe that future designs of MoE routers could be guided by our experiments to bolster their capabilities for certain types of routing tasks. In addition, the fact that MoE routers trained with gradient descent are able to learn meaningful and interpretable clusters is a promising sign for the field of Conditional Compute models as it shows that not only do we save compute but we can also hope to learn a sparse yet interpretable solution to a complex problem.

### 5.1 Open Questions

An important open direction from our work is designing new neural architectures given the insights we uncover about the role of routers in MoE models. We have seen that the standard MoE router struggles in certain settings. This necessitates the design of more complex and capable router modules. Another direction which we believe can help improve our understanding of MoE models further is to extend the theoretical results to more practical settings. For instance, the following would be an interesting question to analyze theoretically. In practice, we don't have a clean separation of concepts or tasks as we did in our theory. Rather, concepts might be organized in a hierarchy or more generally a graph with varying amounts of overlap between nodes. Given such a graph, can we design a router and prove guarantees on it being able to uncover the latent graph structure?

### Limitations

We outline the key limitations of our work here. While we see empirical and theoretical evidence of an MoE router being able to learn intelligent and interpretable routing in many settings, they are brittle in slightly more complex settings such as the dictionary learning setting. We acknowledge that

today's MoE routers perhaps do not learn latent structures always. But we believe there is scope for them to develop the ability to do so in the future. We performed a number of our experiments on carefully crafted synthetic data to ablate clearly the effect of learning to route. However a downside of this is that we lose some amount of generalization of insights to real data settings where the picture might be different. We would like to mention, however, that there is significant evidence in the deep learning field for analyzes such as ours to yield useful insights which generalize to real data and offer actionable ways to improve models. Lastly, some of our experiments were done on reasonably large T5 models which require significant compute.

## Ethics Statement

Our work helps understand one of the key aspects of Mixture of Expert style deep learning models better. There are no direct negative societal or discriminatory impacts of our work. Rather, we hope that the broader impact our work has positive effects for society at large.

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

## A  Additional Background

## B  Additional Related Work

We first discuss comparison of our work with (Chen et al., 2022) in more detail. The focus in Chen et al. (2022) is on designing a data distribution where MoE models have a provable advantage over a non-expert model. In contrast, we are interested in the benefit of MoE models in well-studied clustering instances (such as mixtures of Gaussians) and their variants. In such settings, we prove the first theoretical results showing that MoE models can get very good accuracy. In addition, Chen et al. (2022) focus on convolutional layers, whereas we focus on fully connected layers, which is closer to the standard Transformer architecture. Further, we seek to compare MoE models with dense models and aim to understand the advantage of learned routing vs. non-learned routing. Regarding real-world experiments, our focus is on language data and understanding the benefit of learned routing when used within a transformer architecture (T5X). In contrast, the experiments in Chen et al. (2022) on language data study the benefits of routing when using the representations generated by a large-language model (such as BERT) in a black-box manner.

We discuss some additional related works now. Clark et al. (2022); Tay et al. (2022) established that the benefits of MoE models scale to larger sizes (although the scaling law seems to slow down at extremely large scales). There have been a lot of efforts to study the best way to route tokens which includes ideas from load balancing (Zoph et al., 2022a; Lewis et al., 2021), reinforcement learning (Clark et al., 2022), and smooth tok-$k$ expert selection (Hazimeh et al., 2021). Other choices for the routing function include non-learnable ones such as Locality Sensitive Hashing (Panigrahy et al., 2021), which generally maps similar inputs to the same expert, Hash Layers (Roller et al., 2021) which uses token-based random hashing, and language-specific deterministic routing (Fan et al., 2021). Another approach of Residual Mixture of Experts (Wu et al., 2022) separates the expert weights into input-independent and input-dependent components. MoE ideas have made their impact on prominent large scale systems such as GLAM (Du et al., 2022), GShard (Lepikhin et al., 2020).

Dictionary Learning and Mixture of Gaussians learning problems are well-studied by a long line of works. For references on these problems we refer the reader to the textbook Moitra (2018) and the references within.

## C  Additional Experimental Details

### C.1  Mixture of Gaussians/Subspaces experiments

**Details on the Mixture of Subspaces Setting** In this setting, the input is a mixture of $k$ randomly chosen rank-$t$ subspaces with added noise to samples. For each subspace, let $x$ be a spherical Gaussian restricted to the subspace, and $\epsilon$ is a spherical Gaussian in all dimensions (with a smaller norm than $x$). The examples are generated as $x + \epsilon$, and the label is $w^T \epsilon$ where $w$ is a random vector chosen for the subspace. Note that this is no longer possible with a linear router if we want to route based on the subspace, as the subspaces are not linearly separable. Due to this, we shift to using a depth-3 MLP as the router. There are $k$ experts, and each expert is also a depth-3 MLP. We set $k = 64$ and $t = 8$. The results of our experiments are shown in Figure 6.

**Is the router learning a bipartite matching between clusters and experts?** In Figure 3(b) we saw that each cluster nearly gets routed to a single expert. As each expert is overparameterized it is possible that the router learns to focus on only few experts for all clusters. To verify that this is not the case we compute how uniformly distributed the norms of weights feeding into different experts are (note that the router is linear). Let $M_i$ denote the weights feeding into expert $i$. We compute

$$\frac{\mathbb{E}[\|M_i\|^2]}{\mathbb{E}[\|M_i\|]^2}.$$

This quantity would be 1 if all $\|M_i\|$ were equal and it would be $\#experts$ if only one $\|M_i\|$ was nonzero. In the right most point for the MOE with learned router in Figure 3(a) this quantity is 1.025 while $\#experts$ is 64. Hence it is not the case that the model learns to focus on only a few experts.

**Other details for Figures**

In Figure 2 we trained networks with Adam and a learning rate of 3.2e-4 for the experts and 3.2e-3 for the router.

In Figure 3 and 4 we trained networks with SGD (with .9 momentum) and a learning rate of 1e-2 for the experts and 1e-1 for the router. We choose a higher learning rate for the router as otherwise the *overparameterized* experts may learn to overfit before the expert learns a good routing. The hidden layer width is $4 \cdot 24 = 96$.

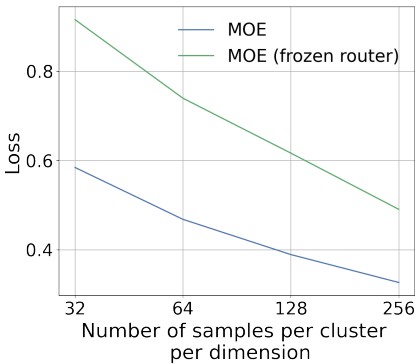

Figure 6: **Mixture-of-subspaces inputs and linear function per subspace.** We see that an MoE model with a learned router outperforms an MoE model with a frozen router across various dataset sizes.

In Figure 6 we trained networks with Adam and a learning rate of 1e-3 for the experts and 3.2e-3 for the router. We choose a higher learning rate for the router as otherwise the *overparameterized* experts may learn to overfit before the expert learns a good routing. The hidden layer width is $4 \cdot 24 = 96$.

## C.2 More Details on Tokenized Variant Experiments

We first describe our setup for the experiments on the tokenized variant of mixture of Gaussians in more detail. We have an equally weighted mixture distribution over 32 Gaussians whose centers are randomly chosen from a 32 dimensional space. The inputs we generate consist of a sequence of tokens where each token is a randomly drawn sample from the mixture. We vary the length of the sequence $s$. The output is one of two functions:

- **Linear Output**: Each Gaussian component in the mixture has an associated vector $w_i$ which yields the final output $y$ as follows

$$y = \sum_{i=1}^{s} w_{g_i}^\top x_i,$$

where $g_i$ is the identity of the cluster of the $i^{th}$ token in the sequence. For most of our experiments in this section, we use this output function.

- **Quadratic Output**: As before, we have $w_i$s corresponding to each Gaussian. The final output is

$$y = \sum_{i=1}^{s} (w_{g_i}^\top x_i)^2,$$

where $g_i$ is the identity of the cluster of the $i^{th}$ token in the sequence. This is a harder setting for a model to solve without figuring out a good routing function.

We experiment with learning the above function using a soft variant of an MoE model which takes in 1 token at a time and routes it to different experts. The expert outputs are weighted using the router softmax probabilities to give the final output. We run experiments with experts which are 1 layer linear functions and compare the performance with a dense MLP model with 2 fully-connected layers. This additional layer allows for a fair comparison as the router can be thought of as a layer in itself.

**Results.** On the linear output function, at small sequence lengths we observe a stronger performance by MoE in comparison to the dense model. As the sequence length increases, we start to see more of the clusters in the mixture present in the input. Since the final output is a linear function of the tokens, a dense model can solve this approximately by learning a single linear function which represents the expected value of the output. We start to see this behavior come out when sequence length increases and the dense model starts to perform just as well as the MoE. We observe that the MoE model attains a non-trivial sparsity and the sparsity is lower than what we get under a shuffled router. On the quadratic output function, we observe that MoE with 32 experts strongly outperform a dense model with the MoE model achieving an MSE loss of 0.0075 compared to an MSE of 0.018 by the dense model. The sparsity measure of the MoE model is 8.8 compared to a shuffled router sparsity of 11.5 showing that the model has learned cluster directions to a non-trivial degree and is using this learnt routing to solve the problem more effectively than a dense model with an additional layer.

| Model \Sequence Length | 4 | 8 | 16 |
|---|---|---|---|
| Dense model with 2 layers | 0.0044 | 0.0021 | 0.0015 |
| MoE with 32 linear experts | 0.0016 | 0.0017 | 0.0015 |
| MoE with 32 linear experts and frozen router | 0.0046 | 0.0020 | 0.0027 |

Table 1: MSE for Tokenized Variant of Mixture of Gaussians experiments.

| Model \Sequence Length | 4 | 8 | 16 |
|---|---|---|---|
| Sparsity | 13 | 14.14 | 14.29 |
| Sparsity with shuffled router (trained) | 14.29 | 15.41 | 15.6 |
| Sparsity with frozen router | 28.52 | 28.51 | 28.53 |

Table 2: Sparsity for Tokenized Variant of Mixture of Gaussians experiments.

### C.3 More Details on Dictionary Learning Experiments

We first describe the setup of our experiments in Section 3.6 in more detail. We sample $c$ random directions in a $d$ dimensional space each corresponding to the center of a spherical Gaussian. We ensure the Gaussians are well-separated from each other. An input $x$ is generated by first sampling from $r$ of the Gaussians (without replacement) and adding them. Here $r$ denotes the rank of the dictionary learning problem and we use $r = 2$ in all our experiments. Additionally, we have a vector $w_i$ associated with each Gaussian which are randomly sampled. For an input $x$ which is generated from Gaussians $i$ and $j$, the output is given by

$$y = (w_i + w_j)^\top x.$$

**Results.** We first run experiments with $c = 32, d = 32$. In this setting, our MoE model uses 32 linear experts. We compare it with a dense model with 2 fully connected layers and a ReLU activation between them. Firstly, we observe that both the dense and the MoE model perform similarly and achieve MSE scores of 0.043 and 0.041 respectively. When we fix the dense model to also be of depth 1, we observe a the MSE increase to 0.19 indicating a larger gap from MoE's performance. However, we don't observe the router learning the cluster center directions. We see a sparsity of 4.5 but the same sparsity is observed even under a shuffled router. This indicates that the MoE model is achieving sparsity in this setting by blowing up the router norm rather than learning useful directions. We do, however, see some evidence that the model is learning something cluster specific. When we measure the average sparsity across samples from different pairs of Gaussians we observe a value of 8.1. This is a measure of how many total experts the model is learning to use. When we fix one of the Gaussian and vary the second one, we observe the average sparsity drop to 6.4. This indicates that on average, when one Gaussian is fixed, the router uses 6.4 experts versus when 8.1 experts used when both Gaussians are varied. To investigate whether this is happening because the problem setting was too easy to solve, we increase $c$ to 128 and keep the number of experts as 32. In this setting we observe the MoE model achieve an MSE of 0.23 as opposed to 0.26

achieved by the dense model. Here, the problem is hard enough that none of the models are able to achieve close to 0 error. Moreover, the sparsity and shuffled sparsity of the MoE model are both 3.01 making it unclear if the MoE model is learning a useful routing. To understand this, we train an MoE model with 32 experts and a frozen router. This achieves a much worst MSE of 0.38 indicating that there is a clear benefit to learning to route, although there is still a lot of room for the MoE model to improve as it is not learning the optimal routing which can fully solve the problem). Overall, we conclude that this setting is a challenging one for the simple form of MoE routing to handle. We leave it open for future work to come up with routing schemes which work well for the dictionary learning setting we propose here.

### C.4 More T5 Experiments with varying number of Experts

We describe the setup of our experiments in Section 3.7 in more detail. We adapt the T5-Base model which is an Transformer encoder-Transformer decoder model with 12 layers each. It uses a model dimension of 768 and hidden dimension of 2048 along with 12 attention heads each of dimension 64. We run pre-training for this model on a language modeling task on the mC4 dataset and measure the next-token prediction accuracy achieved. We use an implementation of the MoE idea in this model which uses 3 sparse layers equally spaced among the 12 layers. It routes to the top-2 experts (instead of top-1) and performs load balancing using an expert capacity factor of 1.25 during training. During evaluation the expert capacity factor is set to 2. We train these models for 500k steps using Adam optimizer.

**Results.** On this model, we wish to study how important it is to learn a good routing function. Even if one freezes the router to its randomly initialized values, due to the earlier layers being allowed to train, the model can implicitly learn clever routing functions due to the large model dimension is has at its disposal. Indeed, we see evidence to support this when we try to learn a Base-sized model with a frozen router, the pre-training accuracy is not affected significantly (70.34% vs 70.27%). To prevent this implicit learning of routing functions from happening we severely restrict the model dimension and hidden dimension of the model. We reduce the model dimension to 32 and hidden dimension to 256 and keep everything else the same. Now we compare training with a frozen router to training with a trainable router and observe a consistent strong improvement in performance when the router is allowed to train (Figure 7) across different total number of experts used.

To add to the experiments in Section 3.7, we perform similar comparisons between the pre-training accuracy of an MoE model trained with frozen routers vs a model with trainable routers at differing total number of experts.

## D Proof of Theorem 1

We first re-state a formal variant of the theorem here.

**Theorem 3.** *Let $f_{b,W,U} : \mathbb{R}^d \to \mathbb{R}^k$ be a prediction function parametrized by weights $b \in \mathbb{R}^h$, $W \in \mathbb{R}^{h \times d}$, and $U$ such that $f_{b,W,U}(x) = g_U(Wx + b)$ for some function $g$. Let $y : \mathbb{R}^d \to \mathbb{R}^k$ be a target function that depends only on some $d' < d$ coordinates; i.e., $y$ is a function of a projection of $x$ onto a $d'$-dimensional subspace $V$. Then any local minimum of the regularized expected loss*

$$L(b, W, U) = \mathbb{E}_x \ell(y, f_{b,W,U}(x)) + \lambda_1 \|W\|^2 + \lambda_2 \|U\|^2,$$
$$\lambda_1, \lambda_2 > 0$$

*must have each row of $W$ within the relevant subspace $V$. That is, for any $i \in [h]$, $Proj_{V^\perp}(W_i) = 0$.*

*Proof.* We prove the theorem for the where $V$ is axis-aligned and is in particular the span of the first $d'$ coordinates; the general case follows by a similar argument once you use rotational symmetry.

We write $x = (x^{(1)}, x^{(2)})$ where $x^{(1)}$ is the first $d' < d$ coordinates of the input $x$. Note that 1) $x^{(1)}$ and $x^{(2)}$ are independent, 2) $x^{(2)}$ is Gaussian.

Consider a local minimum $(b, W, U)$ of $L$. Let $(W_1, W_2)$ denote the column-wise split of $W$ such that $Wx = W_1 x^{(1)} + W_2 x^{(2)}$. For the sake of contradiction assume that $W_2$ is not zero.

Take some $\varepsilon \in (0, 1)$ and define a corresponding $W' = (W_1, W_2') \in R^{h \times d}$ such that $W_2' = (1 - \varepsilon)W_2$. Let $z \in \mathbb{R}^h$ be a Gaussian random vector such that $z$ is independent of $x$ and $W_2' x^{(2)} + z$ equals $W_2 x^{(2)}$

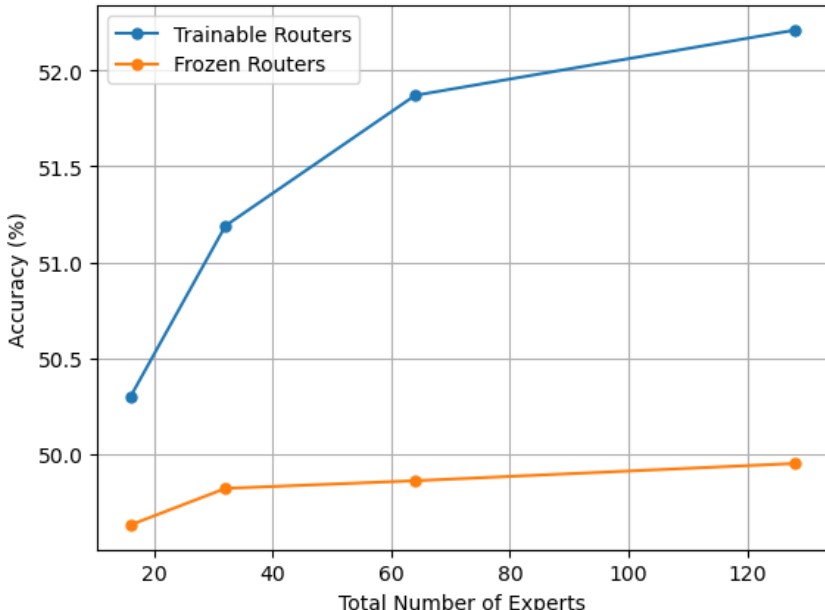

Figure 7: Pretraining Accuracy on mC4 using different T5 models. Each run is a 12 layer model with model dimension reduced to 32 (from 768). This has been done to better capture the role of learning to route. Each model is allowed to train for 500k steps.

in distribution. Since $x^{(2)}$ is Gaussian is suffices that

$$\mathbb{E}[z] = \varepsilon \mathbb{E}[W_2 x^{(2)}],$$
$$\mathrm{Cov}(z) = \sqrt{1 - (1 - \varepsilon)^2} \mathrm{Cov}(W_2 x^{(2)}).$$

Since $x^{(1)}$ and $x^{(2)}$ are independent, we have in fact $(x^{(1)}, W_1 x^{(1)} + W_2' x^{(2)} + z)$ and $(x^{(1)}, W_1 x^{(1)} + W_2 x^{(2)})$ are equal in distribution. We can therefore write

$$\mathbb{E}_x \ell(y, f_{b,W,U}(x^{(1)}, x^{(2)}))$$
$$= \mathbb{E}_{x^{(1)}, x^{(2)}} \ell(y(x), g_U(Wx + b))$$
$$= \mathbb{E}_z \mathbb{E}_{x^{(1)}, x^{(2)}} \ell(y(x^{(1)}), g_U(W_1 x^{(1)} + W_2' x^{(2)} + z + b)).$$

Define

$$z_\star = \arg\min_z \mathbb{E}_{x^{(1)}, x^{(2)}} \ell(y(x^{(1)}), g_U(W_1' x^{(1)} + W_2' x^{(2)} + z + b)).$$

and $b' = z_\star + b$. Then it is clear that

$$\mathbb{E}_x \ell(y, f_{b,W,U}(x^{(1)}, x^{(2)}))$$
$$= \mathbb{E}_z \mathbb{E}_{x^{(1)}, x^{(2)}} \ell(y(x^{(1)}), g_U(W_1 x^{(1)} + W_2' x^{(2)} + z + b))$$
$$\geq \mathbb{E}_{x^{(1)}, x^{(2)}} \ell(y(x^{(1)}), g_U(W_1 x^{(1)} + W_2' x^{(2)} + z_\star + b))$$
$$= \mathbb{E}_{x^{(1)}, x^{(2)}} \ell(y(x^{(1)}), g_U(W'x + b')).$$

and since $\|W\| > \|W'\|$ and $\lambda_1 > 0$,

$$L(b, W, U) > L(b', W', U).$$

Since $\varepsilon$ was arbitrary it follows that $(b, W, U)$ cannot be a local minimum, which is a contradiction. $\qquad\square$

# E  Router Learns Latent Cluster Structure in Mixture of $k$ Gaussians

We begin by describing the problem setting we consider. Let $\{g_c = \mathcal{N}(\mu_c, \sigma_c I_d)\}_{c=1}^k$ be the $k$ spherical component Gaussians of a mixture distribution defined by $\sum_{c=1}^k w_c g_c$. In our work, we assume that all the mixture weights $w_c$ are uniformly $= 1/k$ for simplicity. Let $\{v_c\}_{c=1}^k$ be $k$ cluster specific vectors. Given $x \sim \sum_{i=1}^k g_i/k$, define the ground truth function $f^*(x) = v_{c(x)}$ where $c(x)$ represents the mixture component from which $x$ was sampled.

First, we observe that if $v_c = \mu_c$ for all the components, then the problem has close resemblance to the problem of $k$-means clustering.

**$k$-Means Clustering:** Given input data $\{x\}_{i=1}^n \in \mathbb{R}^d$, the $k$-means objective tries to find $k$ cluster centers $\{u_j\}_{j=1}^k \in \mathbb{R}^d$ such that the average squared distance between each point and its closest cluster center

$$\sum_{i=1}^n \left\| x_i - \sum_{j=1}^k \mathbb{1}\{j = \arg\min_{l \in [k]} \|x_i - u_l\|_2\} u_j \right\|_2^2$$

is minimized.

We present a simplified MoE architecture with weight sharing to formalize this connection with $k$-means.

**MoE Architecture with Weight Sharing:** The neural network is of the form: $f(x) = \sum_{i=1}^e p_i u_i$ where $p_i = 1 \iff i = \arg\min_i \|u_i - x\|_2$ and $u_i$'s are its parameters (experts).

We first note that if we set the ground truth output to be the input $x_i$ in our setting, then the best solution would be $u_j = \mu_j$ which is what would optimize the $k$-means objective as well. Thus $k$-means can be viewed as supervised problem in this sense.

**Lemma 1.** *Gradient Descent on square loss in the above setting with the weight sharing MoE architecture is equivalent to optimizing the $k$-means objective using a step size.*

*Proof.* This loss functions in the two settings match and hence gradient descent will make identical steps. □

Next, we generalize our architecture slightly by dropping the weight sharing assumption to get something closer to an MoE model.

**A more general MoE architecture:** The neural network is of the form: $f(x) = \sum_{i=1}^e p_i u_i$ where $p_i = \text{softmax}(w_i^T x)$ and $u_i, w_i$'s are parameters. The $w_i$'s represent the router and the $u_i$'s represent the experts. Recall that in this setting we have $\{g_i = \mathcal{N}(\mu_c, \sigma_i I_d)\}_{i=1}^k$ as the $k$ spherical component Gaussians of a mixture distribution defined by $\sum_{i=1}^k w_i g_i$. We assume that all the mixture weights $w_i$ are uniformly $= 1/k$ for simplicity. We also assume that the centers $\mu_i$ are all randomly chosen in $d$-dimensional space such that $\|\mu_i\|_2 \leq \alpha\sqrt{d}$. $\{v_i\}_{c=1}^k$ be $k$ cluster specific vectors. Given $x \sim \sum_{i=1}^k g_i/k$, the ground truth function $f^*(x) = v_{c(x)}$ where $c(x)$ represents the mixture component from which $x$ was sampled.

**MoE architecture:** The neural network is of the form: $f(x) = \sum_{i=1}^e p_i u_i$ where $p_i = \text{softmax}(w_i^T x)$ and $u_i, w_i$'s are trainable parameters. The $w_i$'s represent the router and the $u_i$'s represent the experts. Given the above, we restate Theorem 2.

**Theorem.** *Given $n = \Theta(dk \log k)$ samples drawn from a mixture of $k$ spherical Gaussians in $d$-dimensions which are c-separated for some constant c as described above, consider an instantiation of the above MoE architecture with $O(k \log k)$ experts. If we initialize the router weights $w_i$ to randomly drawn examples from the train set, the router will learn to route examples according to the cluster they belong to.*

*Proof.* Recall that $\{g_i = \mathcal{N}(\mu_i, \sigma_i I_d)\}_{i=1}^k$ are the $k$ spherical component Gaussians. Let $X = \{x_i\}_{i=1}^n$ be the train samples and let $W_0 = \{w_j\}_{j=1}^{O(k \log(k))}$ be $O(k \log k)$ randomly selected samples from $X$. We initialize the router weights to $W_0$. A simple coupon collector argument would give us that with

probability $\geq 1 - 1/poly(k)$, there exists at least one sample from each of the $k$ component Gaussians in $W_0$. Moreover, due to the $c$-separated nature of the Gaussians, for any two components $i \neq j$

$$\Pr_{x_i \sim g_i, x_j \sim g_j} \left[ \|x_i - x_j\| \leq C\sqrt{d} \right] \leq \frac{1}{d^{10}}, \tag{1}$$

for a small enough constant $C$. Let $E$ denote the set of all the experts. Note that there is a one-to-one association between an expert $e_j$ and a router weight vector $w_j$. Partition the experts into $k$ sets where $E_k$ is the set of experts whose corresponding router weights were initialized using samples from the $k^{th}$ component $g_k$. We call the set of experts $E_k$ as cluster $k$'s specialists.

When an input $x_i \sim g_k$ needs to be routed, the total probability weight the router assigns to experts not in $E_k$ is $\leq \exp(-O(d))$. Hence, assuming $d \geq k$, the output is going to have contributions largely from the experts in $E_k$. This implies that the contribution of the gradient from the loss on input $x_i$ is minimal on the experts not in $E_k$. Among the experts in $E_k$, there will exist at least one expert which each gradient will push in the direction of $w_k$. This can happen with multiple experts in $E_k$ at the same time as well. Moreover, using the same argument as above, the experts in $E_k$ do not get significantly affected by the gradients coming from examples $x_j$ which are sampled from Gaussians other than $g_k$. By picking a small enough learning rate for the router, this leads to the following configuration after a logarithmic number of gradient descent steps. For every $k$, there will exist an expert in $j \in E_k$ such that $\|u_j - w_k\|_2 \leq o(1)$. In addition, the train loss will also be close to 0. Once this configuration has been reached, the subsequent steps of gradient descent only serve to boost the norms of the router weights without changing their direction by much which leads to a further cementing of the cluster based specialization of the experts. $\qquad \square$

## F  Mixture of two truncated Gaussians

**Task:** We are given inputs drawn from a balanced mixture of two spherical truncated Gaussians in $\mathbb{R}^d$ each with $\sigma I_d$ covariance and means $\pm z \in \mathbb{R}^d$ where $\|z\| = 1$. The Gaussians are truncated at radius $r$ where $r < 1$. The ground truth output is a fixed $\pm v$ for the two clusters respectively.

**Network architecture and optimization setup:** The neural network is of the form: $pu + (1-p)(-u) = (2p - 1)u$ where $p(x) = \text{Sigmoid}(w^T x)$ and $w, u, b$ are parameters. Here $p$ corresponds to the router and $\pm u$ to the 2 experts. We will train with population gradient descent with MSE loss. We initialize the $w$ and $u$ as random vectors with norms tending to 0. We will use learning rates $\eta_u$ and $\eta_w$ for parameters $u$ and $w$ such that $\eta_u \gg \eta_w$.

**Theorem 4.** *Gradient descent on population loss in the above setting will converge to a solution in which $\hat{w} = \pm z, \|w\| \to \infty$ and $u = \pm v$.*

*Proof.* Let $w_0$ denote the initial value of $w$, we assume wlog that $w_0^T z > 0$. Let $G$ and $G'$ denote the distributions of the two truncated Gaussians. The loss is:

$$\mathbb{E}_{x \in G} \|(2p(x) - 1)u - v\|^2 + \mathbb{E}_{x \in G'} \|(2p(x) - 1)u + v\|^2$$

We note that the probability of sampling $x$ under $G$ is the same as probability of sampling $-x$ under $G'$. This allows us to rewrite the above loss as

$$\mathbb{E}_{x \in G} \|(2p(x) - 1)u - v\|^2 + \|(2p(-x) - 1)u + v\|^2$$

Using $p(-x) = 1 - p(x)$ we get that the loss functions is equivalent to

$$L(u, w) = 2\mathbb{E}_{x \in G} \|(q(x)u - v)\|^2$$

where $q(x) = 2p(x) - 1$. As $\eta_u \gg \eta_w$ we can assume assume that throughout the training trajectory $\frac{\partial L}{\partial u} = 0$. Hence

$$\mathbb{E}_{x \in G} 2(q(x)u - v))q(x) = 0$$

solving for $u$ gives us

$$u^*(w) = v \frac{\mathbb{E}_{x \in G} q(x)}{\mathbb{E}_{x \in G} q^2(x)}$$

Now let us consider the partial derivative of loss wrt $w$

$$\frac{\partial L}{\partial w} = 4\mathbb{E}_{x \in G}(q(x)u - v)^T(uq'(x)x)$$
$$= 4\mathbb{E}_{x \in G} q'(x)(q(x)\|u\|^2 - v^T u)z + q'(x)(q(x)\|u\|^2 - v^T u)g$$

By a symmetry argument the second term is of the form $cw$ where $c$ is a scalar. Let us focus on the coefficient of $z$ in first term (omitting constant factors). As $\eta_u \gg \eta_w$ we can assume that $u = u^*(w)$, hence we start by substituting that

$$\mathbb{E}_{x \in G} q'(x)(q(x)\|u\|^2 - v^T u) = \|v\|^2 \mathbb{E}_{x \in G} q'(x)\left(q(x)\left(\frac{\mathbb{E}_{x \in G} q(x)}{\mathbb{E}_{x \in G} q^2(x)}\right)^2 - \frac{\mathbb{E}_{x \in G} q(x)}{\mathbb{E}_{x \in G} q^2(x)}\right)$$

$$= \|v\|^2 \mathbb{E}_{x \in G} q'(x)\left(q(x)\left(\frac{\mathbb{E}_{x \in G} q(x)}{\mathbb{E}_{x \in G} q^2(x)}\right)^2 - \frac{\mathbb{E}_{x \in G} q(x)}{\mathbb{E}_{x \in G} q^2(x)}\right)$$

$$= \|v\|^2 \left(\frac{\mathbb{E}_{x \in G} q(x)}{\mathbb{E}_{x \in G} q^2(x)}\right)\mathbb{E}_{x \in G} q'(x)\left(q(x)\left(\frac{\mathbb{E}_{x \in G} q(x)}{\mathbb{E}_{x \in G} q^2(x)}\right) - 1\right)$$

$$= \|v\|^2 \left(\frac{\mathbb{E}_{x \in G} q(x)}{(\mathbb{E}_{x \in G} q^2(x))^2}\right)(\mathbb{E}_{x \in G}[q'(x)q(x)]\mathbb{E}_{x \in G} q(x) - (\mathbb{E}_{x \in G} q^2(x))(\mathbb{E}_{x \in G} q'(x)))$$

**Claim 1.** *If $w^T z > 0$ then*

$$\mathbb{E}_{x \in G} q(x) > 0$$

.

*Proof.* Let $w = \alpha z + w_\perp$ where $\alpha > 0$ and $w_\perp^T z = 0$. For any point $x = \beta z + x_\perp$ where $x_\perp^T z = 0$ it is easy to see that the point $x' = \beta z - x_\perp$ occurs with the same probability as $x$ in $G$. Hence

$$\mathbb{E}_{x \in G} q(x) = \mathbb{E}_{x \in G}(q(x) + q(x'))/2$$

Let $a = w^T x$ and $b = w^T x'$ then $a + b = 2\alpha z^T z = 2\alpha > 0$. As $h(t) = 1 - 2\text{Sigmoid}(t)$ is anti-symmetric and monotonically increasing function, $a + b > 0$ implies that $h(a) + h(b) > 0$. This in turn implies that $q(x) + q(x') > 0$ as $q(x) = h(a)$ and $q(x') = h(b)$. This is sufficient to prove that

$$\mathbb{E}_{x \in G} q(x) = \mathbb{E}_{x \in G}(q(x) + q(x'))/2 > 0$$

□

**Claim 2.**
$$\mathbb{E}_{x \in G}[q'(x)q(x)]\mathbb{E}_{x \in G} q(x) - (\mathbb{E}_{x \in G} q^2(x))(\mathbb{E}_{x \in G} q'(x)) < 0$$

*Proof.* We will prove the following stronger statement:

$$\mathbb{E}_{x \in G}[q'(x)|q(x)|]\mathbb{E}_{x \in G}|q(x)| < \mathbb{E}_{x \in G}[q^2(x)]\mathbb{E}_{x \in G} q'(x)$$

It is easy to show that $q'(x) = (1 - q^2(x))/2$. Let $\mathcal{D}$ be the distribution of $|q(x)|$ where $x \sim G$. We can equivalently show that

$$\mathbb{E}_{y\in\mathcal{D}}[f(y)y]\mathbb{E}_{y\in\mathcal{D}}y < \mathbb{E}_{y\in\mathcal{D}}[y^2]\mathbb{E}_{y\in\mathcal{D}}f(y)$$

.

where $f(y) = (1 - y^2)/2$. This statement directly follows by applying Lemma 2. $\qquad\square$

This shows us that our gradient update which is proportional to $-\frac{\partial L}{\partial w}$ is of the form $\alpha w + \beta z$ where $\beta > 0$. This implies that GD will converge to a solution which is parallel to $z$. Hence we can restrict ourselves to doing GD with $w$ being parallel to $z$.

Let us rewrite the loss with the constraint that $w = \alpha z$ for some scalar $\alpha$.

$$L(u, w) = 2\mathbb{E}_{x\in G}\|(q(x)u - v)\|^2$$

Denote by $x_1 z$ the component of $x$ along $z$ where $x_1$ is a scalar. Let $G_1$ be the distribution of $x_1$. As $\hat{w} = \pm z$ we know that $q(x)$ is determined by $x_1$. Hence we can rewrite that loss as

$$L(u, \alpha) = 2\mathbb{E}_{x_1\in G_1}\|(q_1(x_1)u - v)\|^2$$

where $q_1(x_1) = q(x_1 z) = 1 - 2\text{Sigmoid}(x_1 w^T z) = 1 - 2\text{Sigmoid}(\alpha x_1)$ as $\|z\| = 1$.

$$
\begin{aligned}
\frac{\partial L}{\partial \alpha} &= 4\mathbb{E}_{x\in G}q_1'(x_1)(x_1)(q_1(x_1)u - v)^T u \\
&= 4\mathbb{E}_{x\in G}\left[q_1'(x_1)(x_1)q_1(x_1)\|u\|^2 - q_1'(x_1)(x_1)v^T u\right] \\
&= 4\|v\|^2\mathbb{E}_{x\in G}\left[q_1'(x_1)(x_1)q_1(x_1)\left(\frac{\mathbb{E}_{x\in G}q(x)}{\mathbb{E}_{x\in G}q^2(x)}\right)^2 - q_1'(x_1)(x_1)\left(\frac{\mathbb{E}_{x\in G}q(x)}{\mathbb{E}_{x\in G}q^2(x)}\right)\right] \\
&= \|v\|^2\left(\frac{\mathbb{E}_{x_1\in G_1}q_1(x_1)}{(\mathbb{E}_{x_1\in G_1}q_1^2(x_1))^2}\right)(\mathbb{E}_{x_1\in G_1}[xq'(x)q(x)]\mathbb{E}_{x\in G}q(x) - (\mathbb{E}_{x\in G}q^2(x))(\mathbb{E}_{x\in G}xq'(x)))
\end{aligned}
$$

Assume w.l.o.g. that $\alpha > 0$, assuming this we will prove that $\frac{\partial L}{\partial \alpha} < 0$ which will be sufficient to prove that as we keep running gradient descent we will get $\hat{w} = z, \|w\| \to \infty$.

As $G$ was a *truncated* Gaussian we have that that $\alpha > 0$ implies that $q_1(x_1) > 0$ for all $x_1 \in G_1$. Hence we only need to prove that

$$\mathbb{E}_{x_1\in G_1}[xq'(x)q(x)]\mathbb{E}_{x\in G}q(x) < (\mathbb{E}_{x\in G}q^2(x))(\mathbb{E}_{x\in G}xq'(x))$$

**Claim 3.** *If $\alpha > 0$ then*

$$\mathbb{E}_{x_1\in G_1}[xq'(x)q(x)]\mathbb{E}_{x\in G}q(x) < (\mathbb{E}_{x\in G}q^2(x))(\mathbb{E}_{x\in G}xq'(x))$$

*Proof.* We will prove the following stronger statement:

$$\mathbb{E}_{x_1\in G_1}[xq_1'(x)|q(x)|]\mathbb{E}_{x\in G}|q(x)| < \mathbb{E}_{x\in G}[q^2(x)]\mathbb{E}_{x\in G}xq_1'(x)$$

It is easy to show that $q'(x) = (1 - q^2(x))/2$ and $x = \ln\left(\frac{1+q(x)}{1-q(x)}\right)/\alpha$. Let $\mathcal{D}$ be the distribution of $|q(x)|$ where $x \sim G$. We can equivalently show that

$$\mathbb{E}_{y\in\mathcal{D}}[f(y)y]\mathbb{E}_{y\in\mathcal{D}}y < \mathbb{E}_{y\in\mathcal{D}}[y^2]\mathbb{E}_{y\in\mathcal{D}}f(y)$$

.

where $f(y) = \ln\left(\frac{1+y}{1-y}\right)(1 - y^2)/(2\alpha)$. Note that $f'(y) = (1 - y\ln\left(\frac{1+y}{1-y}\right))/\alpha$ which is a decreasing function of $y$ for $0 < y < 1$ hence $f$ is concave on $(0, 1)$. Using $f(0) = 0$ and applying concavity on the three points $0, a, b$ where $0 < a < b < 1$ gives us that

$$\frac{f(a)}{a} > \frac{f(b)}{b}$$

.

With this property of $f$ the required statement directly follows by applying Lemma 2 with $T = 1$. $\quad\square$

Hence $w$ will converge to satisfying $\hat{w} = z$ and $\|w\| = \infty$. As we started with truncated Gaussians, we will have $q(x) \to 1$ for all $x \in G_1$ and hence $u = u^*(w) \to v$.

$\square$

We will use the following lemma from (Hidden, 2022), which we reproduce for completeness.

**Lemma 2.** *[ (Hidden, 2022)] Let $f$ be a function which satisfies that for all $0 < a < b < T$ we have*

$$\frac{f(a)}{a} > \frac{f(b)}{b}.$$

*Then for any distribution $\mathcal{D}$ supported only on $(0, T)$ we have*

$$\mathbb{E}[x^2]\mathbb{E}[f(x)] > \mathbb{E}[x]\mathbb{E}[xf(x)]$$

*Proof.* Let $y$ be an independent copy of $x$. We note that

$$
\begin{aligned}
2E[x^2]E[f(x)] - 2E[x]E[xf(x)] &= E[y^2 f(y)] - E[xyf(y)] \\
&\quad + E[y^2 f(x)] - E[yxf(x)] \\
&= E[(x - y)(xf(y) - yf(x))].
\end{aligned}
$$

Because of our assumption the random variable $(x - y)(xf(y) - yf(x)$ is non-negative since $f(y)/y - f(x)/x$ has the same sign as $x - y$.

The desired inequality follows.

$\square$