# OpenReview forum: "On the Benefits of Learning to Route in Mixture-of-Experts Models"
_EMNLP/2023/Conference — EMNLP 2023 Main_

### Official Review · Reviewer_qHJT · 2023-07-25

**Soundness:** 3

**Excitement:**

3: Ambivalent: It has merits (e.g., it reports state-of-the-art results, the idea is nice), but there are key weaknesses (e.g., it describes incremental work), and it can significantly benefit from another round of revision. However, I won't object to accepting it if my co-reviewers champion it.

**Paper Topic And Main Contributions:**

This paper studies the benefits of learning to route in Mixture-of-Experts (MoE) models. The key contributions are:

1. Empirical studies on synthetic and real data showing that learning to route intelligently provides significant benefits over routing randomly or using a fixed routing function. The experiments show the router can identify and exploit latent cluster structure in data.

2. Theoretical analysis proving that with simple MoE architectures and gradient descent, the router provably learns to route inputs from a mixture of Gaussians to the correct expert. This holds under standard assumptions on the mixture components being well-separated.

3. Identification that the benefits of MoE models arise from two factors - having more parameters due to sparsity and the router learning an intelligent routing function. The work provides evidence that the second factor of learned routing is very important.

4. Comparisons between MoE models and dense models. While MoE models outperform smaller dense models, the gap reduces with larger dense models in the overparameterized regime.

5. Insights into what type of cluster structures are easy or hard for simple MoE routers to identify. The router excels at separable mixture components but struggles on more complex latent structures.

**Reasons To Accept:**

1. The paper provides both theoretical analysis and empirical evidence on the advantages of learning to route in Mixture-of-Experts (MoE) models. This sheds light on an important aspect of MoE models that is not yet well understood.

2. Through experiments on synthetic and real data, the paper gives evidence that MoE routers can learn to route inputs intelligently based on latent structure in the data. This is an interesting finding that helps us better understand the behaviors and capabilities of MoE routing mechanisms.

3. The theoretical results prove that MoE models with learned routing can achieve good accuracy on classically studied clustering tasks like mixtures of Gaussians. This provides a theoretical grounding for the empirical advantages of learned routing.

4. The paper introduces several carefully constructed synthetic tasks like mixtures of subspaces, tokenized mixtures of Gaussians etc. that help clearly demonstrate the effects of learned routing. These could be useful testbeds for future work on routing mechanisms.

5. The experiments on real NLP data (language modeling using T5) provide evidence that learned routing gives a significant boost over fixed routing schemes. This validates the relevance of insights from synthetic tasks.

**Reasons To Reject:**

1. The paper lacks experimental results on real-world data. Most of the experiments are on synthetic data specifically designed to demonstrate the benefits of mixture-of-experts models. It would be stronger to show these benefits conclusively on complex, real-world datasets.

2. The theoretical results are limited to simple cases like mixtures of Gaussians and may not extend to more complex data distributions encountered in practice. The assumptions made in the theoretical analysis may not hold in real settings.

3. The paper does not compare to the latest state-of-the-art models. Many advances have been made in sparse expert models since this paper was written. The paper would be stronger if it benchmarked against the current best methods.

4. The writing could be improved for clarity in some areas, especially in describing the theoretical results.

5. The implications of the results are not fully explored. For example, the insights into what the router learns could guide architecture designs, but this direction is not pursued.

**Reproducibility:**

4: Could mostly reproduce the results, but there may be some variation because of sample variance or minor variations in their interpretation of the protocol or method.

**Reviewer Confidence:**

3: Pretty sure, but there's a chance I missed something. Although I have a good feel for this area in general, I did not carefully check the paper's details, e.g., the math, experimental design, or novelty.

---

> ### Author Rebuttal · Authors · 2023-08-29
>
> We thank you for your careful reading of our paper and for the constructive comments. We are glad you found our theoretical and empirical insights valuable. We address your concerns and questions below.
> 1. **Most Experiments are on Synthetic Data**: One of our main goals was to study whether MoE routers learn interpretable functions. This is hard to achieve with real data. This was one of the main reasons we focused so much on synthetic data in our work. We would like to point out that we do study the importance of having a trainable router on real data (the mC4 dataset).
> 1. **Theoretical results tackle simple cases like mixtures of Gaussians**: Firstly, we would like to highlight that mixtures of Gaussians is a powerful framework which, if unrestricted, can model any arbitrary distribution in practice by using the Gaussians as ‘basis’ functions. While we agree it would be nice to have theory for more complex distributions like the ones encountered in practice we want to highlight that it is incredibly difficult to prove theoretical results about more complex distributions in general. Moreover, many theoretical insights in Machine Learning such as the interplay of overparameterization and robustness were obtained by studying Gaussian distributions.
> 1. **Latest SoTA Models**: We would like to highlight that our main motivation in this work was to understand the importance of having a trainable router and to study whether the router learns interpretable functions. Hence, we focused on a clean version of MoE routers which is the most popular variant used today. The SOTA MoE models which exist today (e.g. [1,2,3]), while having some variations over the original MoE paper [4], all use the same fundamental router as a backbone to create different routing decisions.
> 1. **Clarity in Theoretical Results**: We apologize for the lack of clarity in our description of theoretical results. We will undertake the following actions to polish our writing for future versions of the paper.
> 	. We will clean up Section 1.2 focusing more on the implications of our theoretical results rathethan the techniques used to prove them and punt a discussion of the techniques to later in the paper.
> . We will present clearer connections to practice for each of our theoretical results.
> . We will have a discussion of some extensions of our theoretical results that would be interesting and have stronger implications.
> 1. **Implications of our Results for Architecture Design**: This is indeed an intriguing research direction. However, we feel that it is worth a full set of papers on its own as there are many things one could try and it is a challenging task to figure out how to use the insights we gained through our experiments to help design better MoE routers.
>
> - [1] - https://arxiv.org/abs/2202.09368
> - [2] - https://arxiv.org/abs/2202.08906
> - [3] - https://arxiv.org/abs/2308.00951
> - [4] - https://arxiv.org/abs/2101.03961

---

### Official Review · Reviewer_iWp2 · 2023-08-04

**Soundness:** 4

**Excitement:**

3: Ambivalent: It has merits (e.g., it reports state-of-the-art results, the idea is nice), but there are key weaknesses (e.g., it describes incremental work), and it can significantly benefit from another round of revision. However, I won't object to accepting it if my co-reviewers champion it.

**Paper Topic And Main Contributions:**

The paper examines MoE models from an empirical and theoretical perspective, aimed at understanding the value of trainable vs fixed routers and providing some theoretical results for relatively simple mixture of gaussian settings.


**Questions For The Authors:**

Please comment on the second reason to reject- the question of routing utility for large transformers seems to remain unanswered. Can you suggest future work that may be able to address it without artificially constraining the model width?

Regarding Theorem 2, could the theorem hold if the clusters are non-uniform in size?  In a practical use case such as the classification setting is there any utility in stratified sampling or other intelligent sampling of initialization examples to ensure coverage across all the classes?



**Reasons To Accept:**


The paper provides some clear evidence of when a trainable router will outperform a fixed router, and when the benefits of MoEs might be diminished (ie when competing with large dense models).

The work also shows that trainable routers can learn to ignore spurious dimensions in the input.

The theoretical results hint at a good initialization strategy for linear routers.



**Reasons To Reject:**

The contributions are relatively modest and largely limited to simple model architectures.

By artificially constraining the width of the T5 model, the observation that a trainable router outperforms a frozen router seems to be purely academic- the important practical question is whether trainable routing can improve performance for a large (wide) transformer.


**Reproducibility:**

4: Could mostly reproduce the results, but there may be some variation because of sample variance or minor variations in their interpretation of the protocol or method.

**Reviewer Confidence:**

3: Pretty sure, but there's a chance I missed something. Although I have a good feel for this area in general, I did not carefully check the paper's details, e.g., the math, experimental design, or novelty.

---

> ### Author Rebuttal · Authors · 2023-08-29
>
> We thank you for your careful reading of our paper and for the constructive comments. We are glad you found our theoretical and empirical insights valuable. We address your concerns and questions below.
> 1. **Contributions Limited to Simple Model Architectures**: We want to highlight that our paper focuses on the Transformer architecture and the MoE variant of the Transformer architecture. The Transformer architecture is well established to be an incredibly powerful architecture which can scale well to sizes ranging across many orders of magnitudes. Conceptually, apart from the attention component, the Transformer is indeed a simple repetition of a 2-layer MLP architecture which allows us, for the purpose of theoretical study or synthetic data study, to focus on MLPs as an example. Proving theoretical results or developing interpretable results for the Transformer in its entirety turns out to be extremely challenging. Hence, we focused on simpler yet key pieces of the architecture to develop insights.
> 1. **Artificial Constraint on the Width in Experiments**: Thank you for highlighting this point. To clearly ablate out the benefit of having a trainable router, we chose to make the width small. We originally tried freezing the router in a normal width transformer (the T5-Base model). In this setting, we observed that the difference in pre-training accuracy between the model with a frozen vs trainable router was not very large (although there was still an advantage to having a trainable router). As noted in the paper, we hypothesize that this happens due to the frozen router learning implicit routing using the earlier layers in the model.
> Due to this reason it is tricky to ablate out the precise effect of the trainability of the router.
> We would like to point out that recent works ([1,2,3]) in the space of MoE models all use trainable routers even when the routers are very different from each other.
> We believe developing an experiment that could demonstrate the gains of a trainable router in large transformer models is an important yet challenging future direction.
> 1. **Theorem 2 with non-uniform clusters**: Thank you for raising this interesting question. We believe that the results of Theorem 2 can indeed be generalized to hold when the mixture weights are non-uniform as well. This will complicate the proof significantly but there is nothing crucial holding us back from generalizing the theorem to this setting.
> 1. **Utility of Stratified Sampling or Other Intelligent Samplings in practical use cases**: This is an interesting research direction. The use of intelligent samplings in dataset selection is a very actively studied area where we do have some insights such as the following:
> . Selecting a good balanced dataset is important for learning good classifiers
> . Significantly decreasing the dataset size by selecting examples of the right difficulty.
> . Curriculum learning where we present examples in order of increasing difficulty to the model.
> A tangential direction which our work touches upon is that of using data to initialize the parameters of a model. This is known to give benefits in simpler models such as K-means clustering algorithms. For the complex setting of deep neural networks, it is unclear how to initialize a deeper layer using the data embeddings we have at the input layer. However, there is work [4] which shows that attaching an input token based lookup at the middle of a transformer layer has benefits in scaling up the model capacity without adding too much compute. In addition, there has been work which uses the gradients obtained on the data at initialization to sparsify the network ([5,6]).
> As far as we are aware, there isn’t work which uses input samples to smartly initialize the deeper layer routers of an MoE model. We think this is an interesting direction with potential for efficiency savings.
>
> - [1] - https://arxiv.org/abs/2209.15466
> - [2] - https://arxiv.org/abs/2308.00951
> - [3] - https://arxiv.org/abs/2103.16716
> - [4] - https://arxiv.org/abs/2302.00003
> - [5] - https://arxiv.org/abs/2202.12002
> - [6] - https://arxiv.org/abs/1911.13299

---

### Official Review · Reviewer_6A1b · 2023-08-12

**Typos Grammar Style And Presentation Improvements:** No
**Soundness:** 3

**Excitement:**

3: Ambivalent: It has merits (e.g., it reports state-of-the-art results, the idea is nice), but there are key weaknesses (e.g., it describes incremental work), and it can significantly benefit from another round of revision. However, I won't object to accepting it if my co-reviewers champion it.

**Missing References:**

No

**Paper Topic And Main Contributions:**

In summary, the paper explores the broader benefits of MoE models, such as the Switch Transformer, beyond just scaling. It demonstrates that the intelligent routing capabilities of the router lead to advantages in effectively utilizing parameters and partitioning input data. This understanding provides valuable insights into the potential of MoE models in various applications, shedding light on their advantages in both theoretical and practical contexts.
The paper first examines synthetic scenarios where input data is distributed in clusters. Theoretical and empirical analyses demonstrate that the router learns to capture the underlying cluster structure, enhancing its effectiveness. The research then extends to real-world data using the T5X library. It's observed that a trainable router offers a substantial benefit over a non-trainable one, indicating the practical advantages of the learned routing function.

**Questions For The Authors:**

See weaknesses

**Reasons To Accept:**

1. The paper offers theoretical insights into the benefits of Mixture-of-Expert (MoE) models, specifically those like the Switch Transformer. It establishes the advantages of these models beyond mere scalability, which can contribute to the theoretical foundation of the field.

2. The paper is well-structured, presenting its findings and reasoning clearly.

**Reasons To Reject:**

1. The paper does not talks much about the problem setup. The problem is not well motivated.
2. The paper is very difficult to understand to readers new to this field.
3. Paper has some presentation issues as section 1.2 could be made small by just putting key contributions instead of a large section.
4. There is a very small section of problem setup and most of the paper is covered by experiments. However, it would be better if the authors make the presentation better and place things under the correct sections.

**Reproducibility:**

3: Could reproduce the results with some difficulty. The settings of parameters are underspecified or subjectively determined; the training/evaluation data are not widely available.

**Reviewer Confidence:**

2: Willing to defend my evaluation, but it is fairly likely that I missed some details, didn't understand some central points, or can't be sure about the novelty of the work.

---

> ### Author Rebuttal · Authors · 2023-08-29
>
> Thank you for your review and feedback. We are glad you found our theoretical insights valuable. We address your concerns below
> 1. **Limited motivation provided**: We would like to highlight that we devoted almost the entire first page to motivating the specific directions we study in our work. The page limits force us to be selective about the length of the motivation section and we could not delve deeper into how MoE models came to be and what some of the major open questions are in this direction. However, we understand that to a reader new to this field, the motivation provided might be a bit terse. We will add additional background into the problem and prior work in the Appendix in the future versions of our paper.
> 1. **Presentation Issues**: We thank you for highlighting the length of Section 1.2. We will polish this in future versions of the paper. You mentioned that the paper was very difficult to understand to readers new to this field. Could you please highlight which particular parts of the paper were difficult to understand?
>
> Apart the presentation issues, there do not seem to be any technical concerns raised. If you feel our research is technically sound, we urge you to consider raising your soundness score.

---

### Meta-Review · Area_Chair_d9JL · 2023-09-19

**Recommendation:** 5

**Metareview:**

This paper examines MoE models from an empirical and theoretical perspective, aimed at understanding the value of trainable vs fixed routers and providing some theoretical results for relatively simple mixture of gaussian settings.

Pros

It provides theoretical analysis and empirical evidence on the advantages of learning to route in Mixture-of-Experts (MoE) models

 It establishes the advantages of these models beyond mere scalability, which can contribute to the theoretical foundation of the field.

Through experiments on synthetic and real data, the paper gives evidence that MoE routers can learn to route inputs intelligently based on latent structure in the data and that learning to route yields a significant boost over fixed routing schemes.

Cons

Most of the experiments are on synthetic data. It would be more impactful to show its practical usefulness in real-world complex datasets

The authors have done empirical analysis using quite small models (T5-base). In the current dynamics of this space, in order to make this paper really relevant it would be good to understand whether their findings hold true for large enough models.  While for the theoretical part i can understand simplifying the model architecture, but for the empirical analysis its important to show its generalization to larger LLMs.

Presentation of the paper and writing style needs to be improved so that the motivation and the exact problem setup and discussion of the results is clear to readers, especially those who may not be very familiar with the topic

---

### Decision · Program_Chairs · 2023-10-07

**Decision:**

Accept-Main

**Comment:**

This paper examines MoE models from an empirical and theoretical perspective, aimed at understanding the value of trainable vs fixed routers and providing some theoretical results for relatively simple mixture of gaussian settings.

Pros

It provides theoretical analysis and empirical evidence on the advantages of learning to route in Mixture-of-Experts (MoE) models

 It establishes the advantages of these models beyond mere scalability, which can contribute to the theoretical foundation of the field.

Through experiments on synthetic and real data, the paper gives evidence that MoE routers can learn to route inputs intelligently based on latent structure in the data and that learning to route yields a significant boost over fixed routing schemes.

Cons

Most of the experiments are on synthetic data. It would be more impactful to show its practical usefulness in real-world complex datasets

The authors have done empirical analysis using quite small models (T5-base). In the current dynamics of this space, in order to make this paper really relevant it would be good to understand whether their findings hold true for large enough models.  While for the theoretical part i can understand simplifying the model architecture, but for the empirical analysis its important to show its generalization to larger LLMs.

Presentation of the paper and writing style needs to be improved so that the motivation and the exact problem setup and discussion of the results is clear to readers, especially those who may not be very familiar with the topic